# Human-Object Interaction Detection Collaborated with Large Relation-driven Diffusion Models

**Liulei Li**[1], **Wenguan Wang**[2]*, **Yi Yang**[2]

[1]ReLER, AAII, University of Technology Sydney    [2]CCAI, Zhejiang University

https://github.com/0liliulei/DiffusionHOI

## Abstract

Prevalent human-object interaction (HOI) detection approaches typically leverage large-scale visual-linguistic models to help recognize events involving humans and objects. Though promising, models trained via contrastive learning on text-image pairs often neglect mid/low-level visual cues and struggle at compositional reasoning. In response, we introduce DIFFUSIONHOI, a new HOI detector shedding light on text-to-image diffusion models. Unlike the aforementioned models, diffusion models excel in discerning mid/low-level visual concepts as generative models, and possess strong compositionality to handle novel concepts expressed in text inputs. Considering diffusion models usually emphasize instance objects, we first devise an inversion-based strategy to learn the expression of relation patterns between humans and objects in embedding space. These learned relation embeddings then serve as textual prompts, to steer diffusion models generate images that depict specific interactions, and extract HOI-relevant cues from images without heavy fine-tuning. Benefited from above, DIFFUSIONHOI achieves SOTA performance on three datasets under both regular and zero-shot setups.

## 1  Introduction

As a crucial topic in the field of visual scene understanding, human-object interaction (HOI) detection demands not only inferring the semantics and locations of entities but also should comprehend the ongoing events happening between them[1, 2]. Given the complexity and diversity of human activities in object-rich realistic scenes, this task presents challenges in long-tailed distributions and zero-shot discovery[3]. A set of studies seek to tackle these two issues by leveraging large-scale visual-linguistic models (*e.g.*, CLIP[4]) which show strong generalization ability on dozens of tasks. Though strides made, it has been observed that models trained by aligning high-level text-image semantics face difficulties in discerning spatial locations[5], and struggle at compositionality[6] which is a fundamental ability for human to capture new concepts by combining known parts. In fact, both middle-level visual cues (*e.g.*, spatial relation) and compositionality are essential facets for HOI detection. The former can help deduce feasible interactions according to locations between instances, while compositionality contributes significantly to zero-shot generalization. For example, we can easily understand `human-hold-horse` by composing `human-hold-dog` and class `horse` that have encountered previously.

In contrast, the text-to-image diffusion models[7–14] also pre-trained on large-scale image-text pairs, are demonstrating superior capabilities outperforming models like CLIP. Concretely, they are able to generate diverse high-quality images conditioned on textual inputs, showing proficiency in understanding ***high-level semantics***[15, 16]. In addition, the generated images convey reasonable shape, texture, layout, and structure, indicating the comprehension in ***mid/low-level visual concepts*** as generative models[17]. More importantly, the descriptions are typically organized in a compositional

---

*Corresponding Author: Wenguan Wang.

38th Conference on Neural Information Processing Systems (NeurIPS 2024).

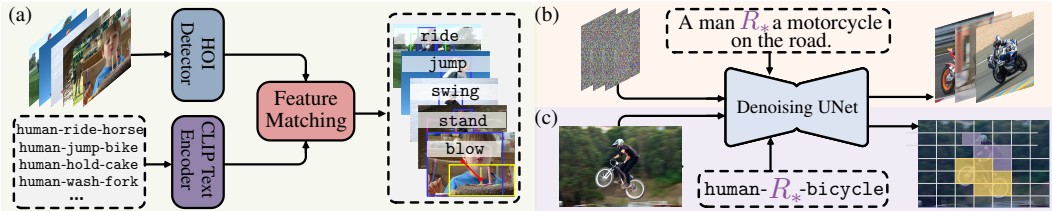

Figure 1: Existing solutions utilize mere linguistic knowledge (a). Our solution utilizes both text-prompt image generation (b) and conditioned feature extraction (c) abilities of diffusion models for knowledge transfer.

manner, with phrases such as "happy","near a bridge", or "hugged by a man" continually appended to objects like "a dog". This suggests that diffusion models inherently possess *compositionality*, to systematically adapt to newly encountered user requirements by composing known visual concepts.

The above analysis motivates us to explore diffusion models for HOI detection. Nonetheless, to fully unlock the potential of diffusion models and accommodate the unique characteristic of HOI detection task, the following questions naturally arise: ❶ With diffusion models typically emphasizing instance generation, how to steer it to prioritize the relationships between humans and objects? ❷ How to transfer the extensive knowledge obtained from large-scale pre-training in diffusion models to assist the recognition of interactions? To address ❶, we harness textual inversion[18] which conceptualizes a user-provided object by inverting it to a text embedding. However, this method focuses solely on instance objects. To facilitate a smooth shift from object-centric to *relation-centric* modeling, we devise a human-object relation inversion strategy grounded in the disentanglement of HOI. Concretely, given the HOI latent describing `human-action-object`, we build a cycle-consistency objective to reconstruct it from a intermediate relation latent derived from the original HOI latent. This reconstruction process is guided by a set of learnable relation embeddings as text prompts, for which we use the placeholder $R_*$ to denote the textual form before encoded into embedding space. These relation embeddings further involves in a relation-centric contrastive learning to enhance the awareness of high-level relational semantics. To answer ❷, we leverage both the text promoted image generation and conditioned feature extraction abilities of diffusion models. We realize *relation-driven* image generation by compositionally organizing $R_*$ with other linguistic elements to formulate new text prompts (Fig.1(b)). This allows for the generation of novel interactions with unseen objects, and extends the training set for HOI detectors. Moreover, we directly utilize diffusion models as backbone to extract HOI-relevant features conditioned on $R_*$ (Fig.1(c)). After a single noise-free forward step, features distinct for each interaction can be obtained. Finally, to establish a loop for mutual boosting between above *relation-inspired* HOI detection and relation modeling, we devise an online update strategy to facilitate the continual evolving of relation embeddings during HOI detection learning.

Benefited from controllable image generation and knowledge transfer from diffusion models, our method named DIFFUSIONHOI enjoys several appealing advantages: **First**, it steers diffusion models to focus on complex relationships rather than single objects in an efficient way. This offers a robust foundation for HOI modeling. **Second**, from the perspective of relation-driven, it unlocks the image generation power of diffusion models tailored for the HOI detection task. This enriches the pool of training samples, particularly for long-tailed/unseen interaction classes. **Third**, the relation-inspired prompting improves both the flexibility and accuracy of HOI detectors. It adapts to each individual image to extract action or object related cues, while CLIP-based methods[3, 19] produce action/object features merely from texts (*i.e*., Fig.1(a)), remaining static and unresponsive to image content.

By embracing text-to-image diffusion models as well as facilitating relation-driven image generation and prompting, our method demonstrates superior performance. It surpasses all top-leading solutions on HICO-DET[20] and V-COCO[21], and sets new state-of-the-arts. In addition, it yields up to **6.43%** mAP improvements on SWiG-HOI[22] under the zero-shot HOI discovery setup. These promising performance evidences the great potential of integrating diffusion models for visual relation understanding. We hope this work could foster the broader exploration of large-scale pre-trained diffusion models on more computer vision tasks beyond mere image generation.

## 2 Related Work

**Human-Object Interaction Detection.** According to the architecture design of networks, existing solutions for HOI detection can be broadly categorized into two groups: one-stage and two-stage.

The one-stage methods[23–26] typically employ a multi-task learning pipeline that jointly undertake the tasks of human-object detection and interaction classification in an end-to-end manner, therefore distinguished by fast inference. In contrast, two-stage methods[27–37] first detect entities with off-the-shelf detectors such as Faster R-CNN[38], and the predict the dense relationships among possible human-object pairs. This paradigm effectively disentangles the HOI detection process and results in improved performance. Inspired by DETR[39], recent advancements shift to adopt Transformer-based architectures[3, 40–46]. Several studies[3, 47–51] also supplement the Transformer-based HOI detectors with large-scale visual-linguistic models like CLIP [4] or visual knowledge[52, 53] to conduct logic-induced reasoning[54]. However, these models focus solely on aligning high-level semantics and overlooking mid/low-level visual cues. To tackle this, we redirect our attention to diffusion models, which perfectly address the aforementioned challenges and possess the capacity to handle previously unseen concepts through their strong compositionality.

**Controllable Image Generation.** To facilitate customized image generation with respect to prede-fined class, attribute, text or image[55], various approaches based on GANs[56] have been proposed. For instance, [57, 58] develop a photo realistic hairstyle transfer method through latent space optimiza-tion. However, these methods typically show limited diversity when compared to likelihood-based models[59]. In response, diffusion models[60–62] have emerged that not only demonstrate remark-able synthesis quality but also offer enhanced controllability. The core idea behind is to transform a simple and known distribution (*e.g.*, Gaussian) into the desired data distribution. These models have proven to be highly effective in various conditional scenarios. According to the conditional targets, the prevalent work can be grouped into class-driven [63, 64] text-driven[7, 8], exemplar image-driven[65, 66], *etc.*. These advances have found application in a wide range of domains such as super resolution[66, 67], image editing[13, 68]. Recently, a new approach achieves guided image generation by learning a single word embedding through a frozen text-to-image model to properly describe the desired target objects[18]. Take inspiration from it, we achieve relation-driven image generation by extending such object-centric concept modeling approach to relation-centric.

**Knowledge Transfer from Diffusion Models.** In light of the notable success achieved by diffusion models in applications, there is a growing interest in transferring knowledge acquired from large-scale pre-training to various tasks[17, 69–75]. For example, given the limited availability of data for constructing NeRFs and the unprecedented generalizability of diffusion models, researchers are motivated to explore generating 3D NeRFs via a 2D text-to-image diffusion model using diverse input text[69–71]. More recently, a notable trend has emerged where efforts are dedicated towards learning semantic representations from diffusion models by extracting intermediate feature maps. It finds diverse application in image segmentation[17], semantic correspondence learning[72–74], and general representation learning[75]. In this work, we extensively harness both the image generation and semantic representation abilities of diffusion models, by using relation-centric embeddings to control the generation and prompt semantic extraction from images with respect to specific interactions.

## 3 Methodology

### 3.1 Preliminary: Textual Inversion

Latent diffusion models [14] represent an evolution of diffusion models which offer significant enhancements in both computational and memory efficiency by executing denosing in the latent space. It comprises two primary components. The first is a pre-trained generator equipped with an encoder $\mathcal{E}$ to map the input image $x$ into a latent vector $z = \mathcal{E}(x)$, from which the original data can be reconstructed via a decoder $\mathcal{D}$ by $\hat{x} = \mathcal{D}(z) \approx x$. The second is a diffusion model to generate latent codes $z$ conditioned on user guidance $y$ which can be text, image, *etc.*The latent codes then serve as inputs to $\mathcal{D}$ for image generation *w.r.t.* $y$. The training objective is given as:

$$\mathcal{L}_{\text{LDM}} := \mathbb{E}_{\mathcal{E}(x),y,\epsilon\sim\mathcal{N}(0,1),t}\Big[\|\epsilon - \epsilon_\theta(z_t, t, c_\theta(y))\|_2^2\Big], \tag{1}$$

where $c_\theta$ is a conditioning model to encode $y$, $z_t$ is the noised latent at time $t$, $\epsilon$ is sampled noise, $\epsilon_\theta$ is the denoising network. Based on latent diffusion models, inversion-based diffusion[18] seeks to learn a text embedding $v_*$ that accurately describes novel concepts in user provided images. This is achieved by optimizing $v_*$ with Eq. 1 to iteratively reconstruct the latent code $z$ of user provided images with text prompts $y$ like "an image of $S_*$", where $S_*$ is the placeholder of new concept:

$$v_* = \arg\min_v \mathbb{E}_{\mathcal{E}(x),y,\epsilon\sim\mathcal{N}(0,1),t}\Big[\|\epsilon - \epsilon_\theta(z_t, t, c_\theta(y))\|_2^2\Big]. \tag{2}$$

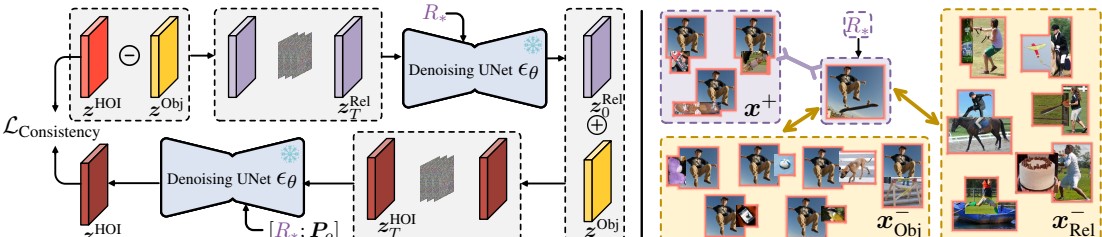

Figure 2: (Left) Disentanglement-based cycle-consistency learning. (Right) Relation-centric contrastive learning.

As such, it enables image generation *w.r.t.* target concepts in diverse scenes by using the learned embedding $v_*$ to replace the tokenized placeholder $S_*$ in text prompts.

### 3.2 Inversion-Based HOI Modeling

**Disentanglement-based Relation Embedding Learning.** To facilitate above inversion technology for relation modeling, two options present: **i)** directly optimizing embeddings describing interactions (*i.e.*, `human-action-object`), which risks overfitting with limited samples for long-tailed categories and cannot generalize to novel concepts, and **ii)** learning `action` embeddings with diverse images sharing a common action but different objects, which seems feasible but poses significant convergence issues due to the complex content, and the optimization target cannot be fixed to actions but not other unrelated elements. In contrast, drawn from the compositional nature of HOI, we adopt a disentangled solution (*i.e.*, Fig. 2) where HOI triplets are broken into `human-action` and `object`. Here `human-action` is considered to describe the relation between `human` and `object`, as `action` is executed by and strictly adheres to `human` involved. Then, denoting the text describing `human-action` as $R_*$, encoded relation embeddings as $v_*^{\text{Rel}} = c_\theta(R_*)$, and the latent of one happening HOI in image as $z^{\text{HOI}}$, a relation latent $z_0^{\text{Rel}}$ could be reconstructed (*i.e.*, denoising with $\epsilon_\theta$ from time $T$ to 0) by:

$$\epsilon_\theta((z^{\text{HOI}} - z^{\text{Obj}})_T, T, v_*^{\text{Rel}}) \to z_0^{\text{Rel}}. \tag{3}$$

Here $(*)_T$ is the noised version at time $T$, and $z^{\text{Obj}}$ is retrieved by encoding the cropped object from image with provided bounding box annotations. We consider $z^{\text{HOI}} - z^{\text{Obj}}$ is able to describe the `human-action` component by subtracting the `object` from `human-action-object`. Then, we can reconstruct the latent representing the complete HOI image by adding $z^{\text{Obj}}$ back to $z_0^{\text{Rel}}$:

$$\epsilon_\theta((z_0^{\text{Rel}} + z^{\text{Obj}})_T, T, [v_*^{\text{Rel}}; P_o]) \to z_0^{\text{HOI}}, \tag{4}$$

where $P_o$ is the CLIP encoded text embedding of `object`, and it is combined with the relation embedding $v_*^{\text{Rel}}$ to generate the prompt that describes the entire HOI image. In this way, with only one learnable relation embedding (*i.e.*, $v_*^{\text{Rel}}$), we build a cycle to generate relation latent $z_0^{\text{Rel}}$ from the HOI image latent $z^{\text{HOI}}$, and subsequently, the original HOI image latent is reconstructed from the generated relation latent. The learning of $v_*^{\text{Rel}}$ can be supervised without human annotation, but just ensuring the consistency between the original HOI latent and the reconstructed one:

$$\mathcal{L}_{\text{Consistency}} = ||\ell_2(z^{\text{HOI}}) - \ell_2(z_0^{\text{HOI}})||_2^2, \tag{5}$$

where all latents are $\ell_2$-normalized for improved training stability [76]. Through such a disentanglement-based relation modeling and cycle-consistency training, the optimization objective become clearer and easier to learn. It enables using same `action` from different interactions to enhance the comprehension of a relation, and generalizing to new interactions by combining it with other `object`.

**Relation-Centric Contrastive Learning.** Eq. 5 is a pixel-level reconstruction loss which prioritizes aligning low-level cues. We supplement it with a relation-centric contrastive loss to enhance the awareness of high-level semantics. Instead of directly engaging learning with relation latents, we combine them with object latents to form new HOI latents, thus significantly enriching the diversity of samples:

$$\begin{aligned} x &= z_0^{\text{Rel}} + z^{\text{Obj}}, & x^+ &= z_0^{\text{Rel}} + p^{\text{Obj}}, \\ x_{\text{Obj}}^- &= z_0^{\text{Rel}} + n_k^{\text{Obj}}, & x_{\text{Rel}}^- &= n_{0,i}^{\text{Rel}} + s_j^{\text{Obj}}, \end{aligned} \tag{6}$$

where $x$ is the anchor sample, $x^+$ is the positive sample composed of a different object latent $p^{\text{Obj}}$ sharing the same class as $z^{\text{Obj}}$. Conversely, $x_{\text{Obj}}^-$ and $x_{\text{Rel}}^-$ are negative samples, with $x_{\text{Obj}}^-$ composed

of a different class object latent $n^{\text{Obj}}$ compared to $x$, and $x_{\text{Rel}}^-$ composed of any other relation latent $n_0^{\text{Rel}}$ and arbitrary object latent $s^{\text{Obj}}$. The final optimization objective is given as:

$$\mathcal{L}_{\text{Contrastive}} = -\log \frac{\exp(\boldsymbol{x} \cdot \boldsymbol{x}^+/\tau)}{\exp(\boldsymbol{x} \cdot \boldsymbol{x}^+/\tau) + \sum_k \exp(\boldsymbol{x} \cdot \boldsymbol{x}_{\text{Obj}}^-/\tau) + \sum_i \sum_j \exp(\boldsymbol{x} \cdot \boldsymbol{x}_{\text{Rel}}^-/\tau)}, \quad (7)$$

to optimize $v_*^{\text{Rel}}$ which involves in reconstructing $\boldsymbol{z}_0^{\text{Rel}}$. $\tau = 0.07$ is the temperature parameter.

### 3.3 Relation-Driven Sample Generation

**Text Prompts Preparation.** We harness the captions provided in the MS COCO Caption dataset[77] to generate diverse prompts. Compared to text synthesized by GPT-4, these captions are more precise and closer to real visual scenes as they are annotated by human subjects. The preparation initiates with a filtration where captions not containing pronouns indicating human (*e.g.*, man, woman, boy) or action words are removed. To further enrich the diversity of prompts, given two randomly selected sentences that share the same action, we exchange the clauses following the action word. Prompts are exclusively generated with GPT-4 only when actions or objects not present in COCO Caption. This results in 33,834 text prompts in total. Finally, action words in prompts are replaced with placeholders corresponding to learned relation embeddings, so as to empower the diffusion model with enhanced awareness of relation patterns between human and object during generation.

**Image and Annotation Generation.** Denoting text prompts as $\mathcal{P} = \{\mathcal{P}_1, \cdots, \mathcal{P}_N\}$, we aim to construct a dataset $\mathcal{X} = \{(\mathcal{I}_1, \mathcal{A}_1), \cdots, (\mathcal{I}_N, \mathcal{A}_N)\}$ where $\mathcal{I}_i \in \mathbb{R}^{H \times W \times 3}$ represents the synthesized image and $\mathcal{A}_i = \{\mathcal{B}_i^h, \mathcal{B}_i^o, \mathcal{C}_i^o, \mathcal{C}_i^a\}$ is the pseudo annotation containing bounding boxes $\mathcal{B}_i^h$ for human, $\mathcal{B}_i^o$ for object, and class labels $\mathcal{C}_i^o$ for object, $\mathcal{C}_i^a$ for action. For the generation of $\mathcal{I}_i$, the text prompts $\mathcal{P}_i$ is first encoded by CLIP text encoder to obtain the conditioning vector $\boldsymbol{P}_i = c_\theta(\mathcal{P}_i) \in \mathbb{R}^d$, where the placeholder string is directly replaced with relation embedding $v_*^{\text{Rel}}$. Then, a random sampled noise tensor $\boldsymbol{z}_T \in \mathbb{R}^{h \times w \times d}$ is iteratively denoised to yield a new latent $\boldsymbol{z}_0$. $\mathcal{I}_i$ is generated by a single pass through $\mathcal{D}$, *i.e.*, $\mathcal{I}_i = \mathcal{D}(\boldsymbol{z}_0)$. For the generation of $\mathcal{A}_i$, $\mathcal{C}_i^o$ and $\mathcal{C}_i^a$ can be easily determined by referring to the action and object words in $\mathcal{P}_i$, while $\mathcal{B}_i^h$ and $\mathcal{B}_i^o$ are derived from the cross-attention maps computed within the U-shape denoising network $\epsilon_\theta$. Specifically, to effectively tackle various input modalities, $\epsilon_\theta$ is equipped with cross-attention mechanisms in each layer to inject $\boldsymbol{P}_i$ into $\boldsymbol{z}$ conforming to the similarity between them. For the $l$-th layer at the last denoising step 0, the cross-attention map is computed as: $\boldsymbol{M}_{i,0}^l = \texttt{softmax}(\boldsymbol{z}_0 \cdot \boldsymbol{P}_i^\top / \sqrt{d}) \in \mathbb{R}^{h \times w}$. According to prior work[17, 78], here $\boldsymbol{M}_{i,0}^l$ signifies the correspondence between text prompt $\mathcal{P}_i$ and regions in generated image. Thus, we explicitly concatenate words describing human and object with $\mathcal{P}_i$ (*i.e.*, $[\mathcal{P}_i; word_{\texttt{human}}; word_{\texttt{object}}]$), resulting in a new text embedding $\hat{\boldsymbol{P}}_i \in \mathbb{R}^{d \times 3}$ and corresponding cross-attention maps $\hat{\boldsymbol{M}}_{i,0}^l \in \mathbb{R}^{h \times w \times 3}$ where the last two items along the third dimension channel are probability maps of human and object. Finally, we leverage the implementation in weakly supervised object localization[79] to outline bounding boxes from these probability maps.

### 3.4 HOI Knowledge Transfer from Diffusion Models

While prior studies[3, 48–50] have investigated knowledge transfer from visual-linguistic models such as CLIP, they utilize visual knowledge solely during training. The prediction relies on a confined set of CLIP encoded word embeddings, which leads to limited knowledge transfer and rigid inference unresponsive to image content. In contrast, we propose directly leveraging diffusion models as the feature extractor and build HOI detector on this basis. Moreover, given the conditioning property of diffusion model, relation embeddings can serve as text prompts to guide the retrieval of interaction-relevant visual cues from images, further benefiting HOI detection.

**HOI Detector Built Upon Diffusion Models.** Pioneering studies[17, 78, 80] have empirically demonstrated that the output of frozen text-to-image diffusion models possesses rich visual features to tackle complex perception tasks. Next we illustrate how to build a HOI detector on this basis. As shown in Fig. 3, our method is a one-stage solution composed of: a visual encoder with diffusion models serving as the backbone, and a HOI decoder consisting of two parallel decoders for instance and interaction detection. For the visual encoder, given an image $\mathcal{I}$, it is encoded into latent space with the encoder $\mathcal{E}$ of a pre-trained generator (*e.g.*, VQGAN): $\boldsymbol{z} = \mathcal{E}(\mathcal{I})$. Then, $\boldsymbol{z}$ is fed into $\epsilon_\theta$ through a single noise-free forward pass to derive text conditioned features: $\epsilon_\theta(\boldsymbol{z}, T, c_\theta(y)) \rightarrow \{\boldsymbol{z}_T^l\}_{l=1}^4$. All

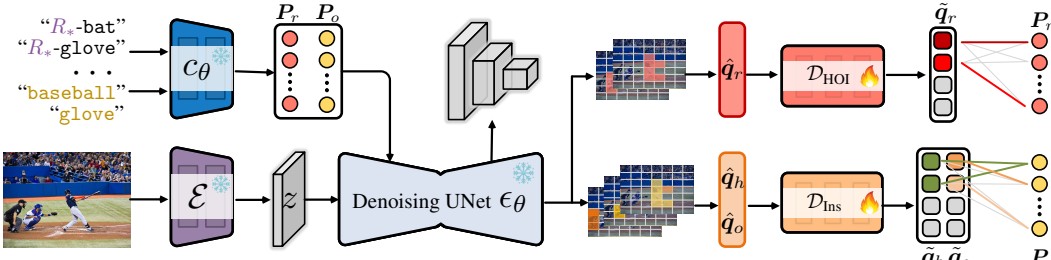

Figure 3: The overall pipeline of DIFFUSIONHOI. See §3.4 for details.

scales of features are aggregated with FPN[81], yielding $z'_T$ in a downsampling factor of 32. The architecture of HOI decoder is similar to GEN-VLKT[3]. Concretely, the instance decoder $\mathcal{D}_{\text{Ins}}$ employs a set of human queries $\{q_h^i\}_{i=1}^{N_q}$ and object queries $\{q_o^j\}_{j=1}^{N_q}$, and considers those at the same index (*i.e.*, $i = j$) as a pair to initialize queries $q_r$ for the interaction decoder $\mathcal{D}_{\text{HOI}}$. In fact, the HOI decoder can be replaced with any other one-stage models. We do not claim the detector architecture as the contribution, but focus on how to derive HOI-relevant feature to assist in HOI detection.

**Relation-Inspired HOI Detection.** As the relation embeddings are optimized towards modeling the interactions between `human` and `object`, we use them as conditions to inspire the extraction of HOI-relevant cues. This can eliminate the potential domain gap between general-purpose diffusion models and the downstream HOI detection task. Specifically, all feasible HOI phrases (*e.g.*, "human feed horse") are encoded with CLIP text encoder into embedding space and concatenated together, with the `human-action` component replaced with learned relation embeddings (*e.g.*, "$R_*$ horse"). This results in HOI prompts $P_r \in \mathbb{R}^{N_r \times d_l}$ which further participates into the cross-attention in $\epsilon_\theta$ via:

$$z_T^l = z_T^l + M_r^l \cdot V_{P_r} \in \mathbb{R}^{h \times w \times d_l}, \quad M_r^l = \texttt{softmax}(z_T^l \cdot K_{P_r}^\top / \sqrt{d}) \in \mathbb{R}^{h \times w \times N_r}, \quad (8)$$

where $K_{P_r}$ and $V_{P_r}$ are key and value embeddings projected from $P_r$. As seen, $P_r$ contributes to: **i)** encourage the denoising network $\epsilon_\theta$ to extract visual features $z_T^l$ *w.r.t.* HOI prompts, and **ii)** guide the derivative of cross-attention maps $M_r^l$ in response to ongoing interactions in $\mathcal{I}$. The final interaction maps are computed as the average value of $\{M_r^l\}_{l=1}^4$. we also derive cross-attention maps for `human` $M_h$ and `object` $M_o$ in a similar way as $M_r$, by without update to $z_T^l$. Then, these cross-attention maps are used to initialize queries from the aggregated visual feature $z'_T$ via mask pooling:

$$\hat{q}_r = \texttt{MaskPooling}(z'_T, M_r^k), \quad \hat{q}_o = \texttt{MaskPooling}(z'_T, M_o), \quad \hat{q}_h = \texttt{MaskPooling}(z'_T, M_h). \quad (9)$$

Note we conduct Hungarian matching between $\hat{q}_r^k$ and $\hat{q}_o^i + \hat{q}_h^j$, so as to arrange HOI, and combined human-object queries that are most similar to the same index in their respective query lists. Following[17], the classification for interaction and instance are jointly supervised by:

$$\begin{aligned} \mathcal{L}_{\text{HOI}} &= \texttt{CE}(\texttt{softmax}(\tilde{q}_r \cdot P_r / \tau_r), y_r) + \texttt{CE}(\texttt{softmax}(\texttt{FFN}(\tilde{q}_r)), y_r), \\ \mathcal{L}_{\text{Ins}} &= \texttt{CE}(\texttt{softmax}(\tilde{q}_o \cdot P_o / \tau_o), y_o) + \texttt{CE}(\texttt{softmax}(\texttt{FFN}(\tilde{q}_o)), y_o), \end{aligned} \quad (10)$$

where $y_r$ and $y_o$ are ground truth for interaction and object categories, $\tilde{q}_r$ and $\tilde{q}_o$ are queries after decoding through $\mathcal{D}_{\text{HOI}}$ and $\mathcal{D}_{\text{Ins}}$, CE and FFN denote the cross entropy loss and feed-forward network. Beyond the score delivered by conventional linear classifier (*i.e.*, $\texttt{softmax}(\texttt{FFN}(\tilde{q}_o))$), here $\texttt{softmax}(\tilde{q} \cdot P / \tau)$ with learnable parameters $\tau_r$ and $\tau_o$ computes the similarity between decoded queries and conditioning prompts, thereby facilitating the recognition for unseen categories.

**Online Update for Relation Embedding.** To enable the continual evolution of relation embeddings $v_*^{\text{Rel}}$ throughout the supervised HOI detection learning, an additional loss considering the compositional nature of HOI is devised. Specifically, we concatenate all $N_a$ relation embeddings into a new prompt $P_a$, from which a set of relation query embeddings $\hat{q}_a$ can be initialized in the same way as $\hat{q}_o$ (*c.f.*, Eq.9). In addition, another set of embeddings describing relations can be derived from $\tilde{q}_r$ and $\tilde{q}_o$ by: $\tilde{q}_a = \tilde{q}_r - \tilde{q}_o$. The goal is to align $\hat{q}_a$ directly derived from visual features with relation embeddings as conditions, and $\tilde{q}_a$ computed from interaction and object queries after decoding:

$$\mathcal{L}_{\text{Rel}} = ||\ell_2(\hat{q}_a) - \ell_2(\tilde{q}_a)||_2^2. \quad (11)$$

Here $\mathcal{L}_{\text{Rel}}$ solely optimizes $v_*^{\text{Rel}}$ to render a mutual boost between HOI detection and relation embedding learning. Concretely, enhanced relation embeddings inspires improved HOI feature discovery, and in turn, the more precise query decoding benefits the update of relation embeddings.

Table 1: Quantitative results for regular HOI detection on HICO-DET[20] and V-COCO[21].

| Method | Backbone | VL Pretrain | Default | | | Known Object | | | V-COCO | |
|---|---|---|---|---|---|---|---|---|---|---|
| | | | Full | Rare | Non-Rare | Full | Rare | Non-Rare | $AP_{role}^{S1}$ | $AP_{role}^{S2}$ |
| iCAN[83] [BMVC18] | R50 | - | 14.84 | 10.45 | 16.150 | 16.26 | 11.33 | 17.73 | 45.3 | - |
| PPDM[24] [CVPR20] | HG104 | - | 21.73 | 13.78 | 24.10 | 24.58 | 16.65 | 26.84 | - | - |
| HOTR[41] [CVPR21] | R50 | - | 23.46 | 16.21 | 25.60 | - | - | - | 55.2 | 64.4 |
| QPIC[42] [CVPR21] | R101 | - | 29.90 | 23.92 | 31.69 | 32.38 | 26.06 | 34.27 | 58.3 | 60.7 |
| CDN[44] [NeurIPS21] | R101 | - | 32.07 | 27.19 | 33.53 | 34.79 | 29.48 | 36.38 | 63.9 | 65.9 |
| CPChoi[84] [CVPR22] | R50 | - | 29.63 | 23.14 | 31.57 | - | - | - | 63.1 | 65.4 |
| STIP[85] [CVPR22] | R50 | - | 32.22 | 28.15 | 33.43 | 35.29 | 31.43 | 36.45 | 66.0 | 70.7 |
| UPT[86] [CVPR22] | R101 | - | 32.62 | 28.62 | 33.81 | 36.08 | 31.41 | 37.47 | 61.3 | 67.1 |
| Iwin[87] [ECCV22] | R101 | - | 32.79 | 27.84 | 35.40 | 35.84 | 28.74 | 36.09 | 60.9 | - |
| MCPC[88] [ECCV22] | R50 | - | 35.15 | 33.71 | 35.58 | 37.56 | 35.87 | 38.06 | 63.0 | 65.1 |
| PViC[89] [ICCV23] | R50 | - | 34.69 | 32.14 | 35.45 | 38.14 | 35.38 | 38.97 | 62.8 | 67.8 |
| PViC†[89] [ICCV23] | Swin-L | - | 44.32 | 44.61 | 44.24 | 47.81 | 48.38 | 47.64 | 64.1 | 70.2 |
| GEN-VLK[3] [CVPR22] | R101 | CLIP | 34.95 | 31.18 | 36.08 | 38.22 | 34.36 | 39.37 | 63.6 | 65.9 |
| HOICLIP[19] [CVPR23] | R50 | CLIP | 34.69 | 31.12 | 35.74 | 37.61 | 34.47 | 38.54 | 63.5 | 64.8 |
| CQL[90] [CVPR23] | R50 | CLIP | 35.36 | 32.97 | 36.07 | 38.43 | 34.85 | 39.50 | 66.4 | 69.2 |
| ViPLO[91] [CVPR23] | ViT-B | CLIP | 37.22 | 35.45 | 37.75 | 40.61 | 38.82 | 41.15 | 62.2 | 68.0 |
| AGER[92] [ICCV23] | R50 | CLIP | 36.75 | 33.53 | 37.71 | 39.84 | 35.58 | 40.2 | 65.7 | 69.7 |
| RmLR[93] [ICCV23] | R101 | MobileBERT | 37.41 | 28.81 | 39.97 | 38.69 | 31.27 | 40.91 | 64.2 | 70.2 |
| ADA-CM[94] [ICCV23] | ViT-L | CLIP | 38.40 | 37.52 | 38.66 | - | - | - | 58.6 | 64.0 |
| DIFFUSIONHOI | VQGAN | Stable Diffusion | **38.12** | **38.93** | **37.84** | **40.93** | **42.87** | **40.04** | **66.8** | **70.9** |
| DIFFUSIONHOI | ViT-L | Stable unCLIP | **42.54** | **42.95** | **42.35** | **44.91** | **45.18** | **44.83** | **67.1** | **71.1** |

†: Models built upon advanced object dector, *i.e.*, $\mathcal{H}$-Deform-DETR[95].

## 3.5 Implementation Details

**Network Architecture.** DIFFUSIONHOI is built upon Stable Diffusion v1.5 with xFormers[82] installed. The denoising UNet $\epsilon_\theta$ receives input latents at a downsampling factor of 1/8, with four encoder blocks output feature at a size of $1/2^{l+3}$ where $l$ is the block index. For the final visual feature $z'$ after FPN aggregation, it is interpolated to a size of 1/32 and then projected to 256 channels to enhance computing efficiency. Both $\mathcal{D}_{HOI}$ and $\mathcal{D}_{Ins}$ consist of six Transformer decoding layers with hidden dimension of 768. The query number $N^q$ is uniformly set to 64 for both $\mathcal{D}_{HOI}$ and $\mathcal{D}_{Ins}$.

**Training Objective.** The inversion-based HOI modeling is jointly optimized by two embedding learning losses: $\mathcal{L}_{Inversion} = \mathcal{L}_{Consistency} + \lambda_1 \mathcal{L}_{Contrastive}$, where $\lambda_1 \in [0, 0.2]$ is scheduled following a cosine annealing policy. For HOI detection learning, we follow DETR[39] to match predictions and ground truths with Hungarian algorithm. Denoting the bounding box detection loss as $\mathcal{L}_{Det}$, the final training objective is given as: $\mathcal{L} = \mathcal{L}_{HOI} + \mathcal{L}_{Ins} + \mathcal{L}_{Det} + \lambda_2 \mathcal{L}_{Rel}$ where $\lambda_2$ is fixed to 0.5.

# 4 Experiment

## 4.1 Experimental Setup

**Datasets.** We conduct extensive experiments on three datasets.

- HICO-DET[20] is a large-scale HOI detection benchmark with 38,118/9,658 images for training/testing, respectively. This dataset includes 80 object categories as in MS-COCO[96] and 117 action categories, formulating a rich vocabulary of 600 human-object interactions in total.
- V-COCO[21] is a curated subset of MS-COCO[96] including 2,533/2,867/4,946 images in `train`/`val`/ `test` sets. It also contains 80 object categories from MS-COCO[96] and a much smaller set of 29 action classes, resulting in a total of 263 human-object interactions.
- SWiG-HOI[22] is assembled from SWiG[97] and DOH[98] with about 45,000/14,000 for training/testing. This dataset covers 406 human actions and 1,000 object categories.

**Zero-Shot HOI Discovery.** In accordance with prior research[3, 19, 99–102], the zero-shot HOI discovery on HICO-DET[20] uses four setups: Rare First Unseen Combination (RF-UC), Non-rare First Unseen Combination (NF-UC), Unseen Verb (UV), and Unseen Object (UO). The RF-UC and NF-UC configurations excluded the 120 most frequent/infrequent interaction categories from the training sets for testing purposes only. The UV and UO setups reserve 20 verb classes and 12 object classes never encountered during training for testing. For SWiG-HOI[22], the test set includes approximately 5,500 interactions, with around 1,800 of them not present in the training set.

Table 2: Zero-shot generalization on HICO-DET [20].

| Method | Type | Unseen | Seen | Full |
|---|---|---|---|---|
| ATL[101] [CVPR21] | RF-UC | 9.18 | 24.67 | 21.57 |
| FCL[100] [CVPR21] | RF-UC | 13.16 | 24.23 | 22.01 |
| SCL[102] [ECCV22] | RF-UC | 19.07 | 30.39 | 28.08 |
| GEN-VLKT[3] [CVPR22] | RF-UC | 21.36 | 32.91 | 30.56 |
| OpenCat[103] [CVPR23] | RF-UC | 21.46 | 33.86 | 31.38 |
| HOICLIP[19] [CVPR23] | RF-UC | 25.53 | 34.85 | 32.99 |
| DIFFUSIONHOI | RF-UC | **32.06** | **36.77** | **35.89** |
| ATL[101] [CVPR21] | NF-UC | 18.25 | 18.78 | 18.67 |
| FCL[100] [CVPR21] | NF-UC | 18.66 | 19.55 | 19.37 |
| SCL[102] [ECCV22] | NF-UC | 21.73 | 25.00 | 24.34 |
| GEN-VLKT[3] [CVPR22] | NF-UC | 25.05 | 23.38 | 23.71 |
| OpenCat[103] [CVPR23] | NF-UC | 23.25 | 28.04 | 27.08 |
| HOICLIP[19] [CVPR23] | NF-UC | 26.39 | 28.10 | 27.75 |
| DIFFUSIONHOI | NF-UC | **30.04** | **30.29** | **30.25** |
| ATL[101] [CVPR21] | UO | 5.05 | 14.69 | 13.08 |
| FCL[100] [CVPR21] | UO | 15.54 | 20.74 | 19.87 |
| GEN-VLKT[3] [CVPR22] | UO | 10.51 | 28.92 | 25.63 |
| OpenCat[103] [CVPR23] | UO | 23.84 | 28.49 | 27.72 |
| HOICLIP[19] [CVPR23] | UO | 16.20 | 30.99 | 28.53 |
| DIFFUSIONHOI | UO | 22.37 | **32.03** | **31.12** |
| GEN-VLKT[3] [CVPR22] | UV | 20.96 | 30.23 | 28.74 |
| HOICLIP[19] [CVPR23] | UV | 24.30 | 32.19 | 31.09 |
| DIFFUSIONHOI | UV | **28.05** | **33.24** | **32.67** |

Table 3: Zero-shot generalization on SWiG-DET [22].

| Method | Non-rare | Rare | Unseen | Full |
|---|---|---|---|---|
| QPIC[42] [CVPR21] | 16.95 | 10.84 | 6.21 | 11.12 |
| THID[48] [CVPR22] | 17.67 | 12.82 | 10.04 | 13.26 |
| CMD-SE[104] [CVPR24] | 21.46 | 14.64 | 10.70 | 15.26 |
| DIFFUSIONHOI | **25.59** | **20.61** | **18.93** | **21.69** |

Table 4: Comparison of parameters and running efficiency. * means applying accelerated technology.

| Method | Backbone | Trainable Params (M) | FPS | HICO-DET |
|---|---|---|---|---|
| Two-stages Detectors: | | | | |
| iCAN[83] [BMVC18] | R50 | 39.8 | 6.23 | 14.84 |
| DRG[105] [ECCV20] | R50 | 46.1 | 6.05 | 19.26 |
| STIP[85] [CVPR22] | R50 | 50.4 | 7.12 | 32.22 |
| ViPLO[91] [CVPR23] | ViT-B | 118.2 | 5.66 | 37.22 |
| ADA-CM[94] [ICCV23] | ViT-L | 6.6 | 3.24 | 38.40 |
| One-stages Detectors: | | | | |
| PPDM[24] [CVPR20] | HG104 | 194.9 | 17.58 | 21.73 |
| HOTR[41] [CVPR21] | R50 | 51.2 | 15.92 | 23.46 |
| QPIC[42] [CVPR21] | R50 | 41.9 | 17.41 | 29.07 |
| CDN[44] [NeurIPS21] | R50 | 42.1 | 16.24 | 31.78 |
| GEN-VLKT[3] [CVPR22] | R50 | 42.8 | 18.23 | 33.75 |
| DIFFUSIONHOI | VQ-GAN | 27.6 | 9.49 | 38.12 |
| *DIFFUSIONHOI | VQ-GAN | 27.6 | 24.77 | 38.12 |

**Evaluation Metric.** Following conventions[3, 41, 42], we adopt mAP as metrics. For HICO-DET, we report performance according to Default and Known Object two setups. The former computes mAP across all testing images, while the latter is tailored for each object class. For each setup, the scores are reported in Full/Rare/Non-Rare three types. For V-COCO, we evaluate the performance under scenario 1 (S1) which contains all 29 actions and scenario 2 (S2) which excludes 4 actions interact with no objects. For zero-shot setup, the evaluation is divided into Seen/Unseen/Full three sets for HICO-DET, and Non-Rare/Rare/Unseen/Full four sets for SWiG-HOI.

**Training and Testing.** The diffusion model and CLIP text encoder are kept frozen during training. For inversion-based HOI modeling, the only learnable parameters are relation embeddings, which are updated for 40,000 steps using images sampled from HICO-DET. Following [18], we employ a base learning rate of $8e^{-2}$ with a batch size of 32. For HOI detection learning, we train the interaction decoder $\mathcal{D}_{Ins}$ and object decoder $\mathcal{D}_{HOI}$ for 60 epochs with a base learning rate of $1e^{-4}$ and batch size of 16, using both synthesized data and the target dataset. Subsequently, the model is trained only on the target dataset for an additional 30 epochs with a base learning rate of $1e^{-5}$. During inference, no data augmentation is used to ensure fair comparison. Following [3, 103], the inputs are resized to maximum of 1,333 pixels on long sides, and the shortest sides falls between 480 and 800 pixels.

**Reproducibility.** DIFFUSIONHOI is implemented in PyTorch and trained on 8 Tesla A40 GPUs with 48GB memory per card.

## 4.2 Comparison with State-of-the-Arts

**Regular Setup.** We first compare DIFFUSIONHOI with top-leading solutions on HICO-DET[20] and V-COCO[21] under the regular setup. As shown in Table 1, for HICO-DET, our method achieves the best performance on both Default and Known Object setups. Notably, with the encoder of VQGAN as the backbone, it surpasses the previous SOTA, RemLR[93], which employs a similar level backbone (*i.e.*, ResNet-50) by **1.19%** and **2.64%** on the Full categories. Benefited from synthesized data and comprehensive knowledge transfer from diffusion models, the performance on Rare categories improves significantly, achieving higher scores than on Non-Rare categories for the first time. Finally, with a more powerful VL model (*i.e.*, Stable unCLIP[11]) and backbone (*i.e.*, ViT-L), the performance is boosted to **42.54%** under the Default setup, surpassing nearly all existing work by a considerable margin. Please note that PViC[89] with Swin-L as the backbone leverages $\mathcal{H}$-Deform-DETR[95] as the detector which achieves 48.7 mAP on MS COCO[96] by running merely 12 epochs, significantly higher than DETR[39] which achieves 36.2 mAP by running 50 epochs.

**Zero-Shot Setup.** Next we investigate the effectiveness of DIFFUSIONHOI under the zero-shot generalization setup. As shown in Table 2, our method yields remarkable performance across all four setups on HICO-DET. In particular, it surpasses the previous SOTA (*i.e.*, HOICLIP[19]) by

Table 5: Detailed analysis of essential components of DIFFUSIONHOI on HICO-DET[20].

| Algorithm Component | Default | | | RF-UC | | |
|---|---|---|---|---|---|---|
| | Full | Rare | Non-Rare | Full | Unseen | Seen |
| BASELINE | 33.24 | 30.25 | 34.32 | 30.47 | 20.63 | 33.09 |
| + Synthesized Data *only* | 35.49 | 36.27 | 35.02 | 33.79 | 28.22 | 34.85 |
| + Relation Prompting *only* | 36.45 | 35.78 | 36.71 | 34.25 | 26.57 | 35.58 |
| DIFFUSIONHOI | 38.12 | 38.93 | 37.84 | 35.89 | 32.06 | 36.77 |

Table 7: Analysis of relation embeddings with different learning strategies for relation-inspired HOI detection.

| Learning Strategy | Default | | | RF-UC | | |
|---|---|---|---|---|---|---|
| | Full | Rare | Non-Rare | Full | Unseen | Seen |
| Textual Inversion | 34.03 | 32.17 | 34.61 | 30.93 | 21.55 | 33.45 |
| Cycle-Consistency | 35.23 | 34.56 | 35.46 | 32.96 | 24.24 | 34.54 |
| + Relation-Centric CL | 35.94 | 35.06 | 36.32 | 33.72 | 25.73 | 35.17 |
| + Online Update | 36.45 | 35.78 | 36.71 | 34.25 | 26.57 | 35.58 |

Table 6: Analysis of conditioning input for relation-inspired HOI detection on HICO-DET[20].

| Conditioning Input | Default | | | RF-UC | | |
|---|---|---|---|---|---|---|
| | Full | Rare | Non-Rare | Full | Unseen | Seen |
| - | 33.24 | 30.25 | 34.32 | 30.47 | 20.63 | 33.09 |
| Textual Description | 33.71 | 30.98 | 34.73 | 30.72 | 21.29 | 33.24 |
| Relation Embedding | 36.45 | 35.78 | 36.71 | 34.25 | 26.57 | 35.58 |

Table 8: Analysis of prompts for dataset generation. *TD*: textual description, *RE*: relation embedding.

| Training Set | Default | | | RF-UC | | |
|---|---|---|---|---|---|---|
| | Full | Rare | Non-Rare | Full | Unseen | Seen |
| HICO-DET | 33.24 | 30.25 | 34.32 | 30.47 | 20.63 | 33.09 |
| + *TD* Synthesized Data | 32.54 | 30.04 | 33.49 | 30.12 | 20.55 | 32.57 |
| + *RE* Synthesized Data | 35.49 | 36.27 | 35.02 | 33.79 | 28.22 | 34.85 |

**2.90%** under the RF-UC setup. This setup emphasizes compositional generalization which requires models to comprehend new types of interactions using known actions and objects. It aligns well with the strengths of text-to-image diffusion models to generate images conditioned on compositionally organized textual descriptions. Moreover, due to the effective knowledge transfer, DIFFUSIONHOI also achieves satisfactory improvement under the UV and UO setups which focus on the recognition of novel actions and objects. Table 3 further confirms the exceptional ability of our method, showing **5.97%/8.23%** mAP improvements over CMD-SE[104] under Rare and Unseen two categories.

**Model Efficiency.** We compare the trainable parameter number and inference time in Table 4. As seen, DIFFUSIONHOI demonstrates significantly fewer trainable parameters compared to the one-stage counterparts. This is attributed to our inversion-based HOI modeling, which avoids fine-tuning diffusion models like previous work[78], while effectively capturing task-specific properties. Regarding inference speed, even with stable diffusion for feature extraction, our method still achieves 9.49 FPS, a rate similar to two-stage models. This is due to the inference involving only one single forward pass, and the downsampling factor of stable diffusion from 1/8 to 1/64 is smaller than conventional backbones typically from 1/4 to 1/32. Moreover, thank to the flourishing community of stable diffusion, a variety of optimized inference solutions have emerged. By running at fp16 precision and using traced UNet, the FPS increases to 24.77, surpassing most one-stage methods.

## 4.3 Diagnostic Analysis

**Key Component Analysis.** We first examine the essential components of DIFFUSIONHOI in Table 5. Here BASELINE denotes HOI detector built upon stable diffusion without text prompting. Through jointly training with the synthesized data, both Default and RF-UC setups observe notable improvements (*e.g.*, up to **2.25%** and **3.32%** on Full categories ). This verifies the effectiveness of our relation-driven HOI image generation strategy. in addition, after imposing relation embedding to prompt the feature extraction and HOI detection processes, the performance boosts to **36.45%** and **34.25%** under two setups. Finally, after combining these two core components together, our DIFFUSIONHOI delivers consistent improvements and sets new SOTA across all setups.

**Conditioning Input.** To assess the effectiveness of learned relational embeddings, we present the experimental results using different conditional inputs to stimulate HOI detection in Table 6. As seen, though action words offer limited improvement, they are far surpassed by relation embeddings which enables HOI-oriented feature extraction and enhance query initialization through cross-attention.

**Relation Embedding Learning.** Next we probe the impact of different strategies for relation embedding learning. The results regarding relation-inspired HOI detection are summarized in Table 7. It can be observed that textual inversion, which directly uses different images sharing the same action for relation embedding learning, is inferior to our cycle-consistency learning strategy that considers the disentanglement nature of HOI interactions. On this basis, the relation-centric contrastive learning and online update strategies consistently bring improvement in both setups.

**Prompt for Dataset Generation.** Finally we study the impact of data synthesized by different types of textual prompts in Table 8. As observed, data generated with purely textual description using plain action words like "The man at bat readies to swing at the pitch" gives negative improvement over baseline. The potentially indicates that diffusion models cannot understand the relations between human-object pairs and generate meaningful images when provided with straightforward textual

Table 9: Comparison of total training time on HICO-DET[20].

| Method | CDN[44] | HOTR[41] | UPT[86] | STIP[85] | GEN-VLKT[3] | HOICLIP[19] | CQL[90] | DIFFUSIONHOI |
|---|---|---|---|---|---|---|---|---|
| Time (Hour) | 25.2 | 23.6 | 17.9 | 16.4 | 28.4 | 29.1 | 29.7 | **5.7+11.5** |

description. In contrast, through relation modeling, data generated with relation embeddings to replace the plain action words provides high-quality samples for the training of HOI detectors.

**Analysis on Training Cost.** For our inversion-based HOI modeling to learn relation-centric embeddings, unlike the original textual inversion technology that learns text embeddings within the image space, we optimize relation embeddings within the latent space by reconstructing interaction features. This lead to reduced training costs. Consequently, the 117 relation embeddings in HICO-DET[20] can be learned within **5.7** hours (23 minutes per relation embedding) which is more efficient than textual inversion (*i.e.*, 32 minutes per embedding). For the main training of HOI detection on HICO-DET, since our method utilizes significantly fewer trainable parameters compared to existing work (*e.g.*, 27.6M *v.s.* 50.4M for STIP[85], 41.9M for QPIC[42], and 42.8M for GEN-VLKT[3] in Table 4), the training process can be completed in just **11.5** hours. The comparison of the whole training time with some representative work is summarized in Table 9, with all experiments conducted on 8 Tesla A40 cards. It can be observed that DIFFUSIONHOI requires less training time than most existing work.

## 5    Conclusion

We present DIFFUSIONHOI, a new HOI detector built upon diffusion models. By explicitly modeling the relations between humans and objects in an inversion-based manner, we enable effective knowledge transfer from diffusion models while adapting unique characteristics of the HOI detection task. This is achieved in two aspects: **i)** *relation-driven* image generation using diffusion models to enrich the training set with more HOI-oriented samples, and **ii)** *relation-inspired* HOI detection with learned relation embeddings as prompts to retrieve task-specific features from images, thereby enhancing the recognition of ongoing interactions. Extensive experiments demonstrate that DIFFUSIONHOI excels in both regular or zero-shot setups and sets new SOTAs. We believe this work provides insights to unleash the power of diffusion models for downstream visual perception tasks in an efficient manner.

**Acknowledgement.** This work was supported by the National Science and Technology Major Project (No. 2023ZD0121300), the National Natural Science Foundation of China (No. 62372405), the Fundamental Research Funds for the Central Universities 226-2024-00058, National Key Laboratory of Human-Machine Hybrid Augmented Intelligence, Xi'an Jiaotong University (No. HMHAI-202403), and CIPSC-SMP-Zhipu Large Model Cross-Disciplinary Fund.

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

# A  Discussion

## A.1  Limitation

One potential limitation of our method is that it does not run as quickly as previous one-stage methods due to the adoption of diffusion models. A detailed comparison is provided in Table 4. As shown, DIFFUSIONHOI achieves inference speeds similar to two-stage methods. However, currently there is a trend towards leveraging large-scale pre-trained models to assist in various perception tasks. These models offer substantial enhancements in performance and accuracy. For instance, after utilizing diffusion models, our method has demonstrated an 8.23% improvement on unseen categories in the SWiG-HOI benchmark. Though it is inevitable that these advancements introduce additional computational overhead, the trade-off is generally considered acceptable given the significant improvements in task performance and the ability to generalize better to previously unseen scenarios. Therefore, an important consideration in future research is the balance between computational cost and the enhanced capabilities provided by large-scale pre-trained models. In this work, we implement acceleration technology for Stable Diffusion which successfully boosts the inference speed of DIFFUSIONHOI to 24.77 FPS, surpassing most one-stage methods.

## A.2  Failure Case

As shown in Fig. S1, we found that failure cases primarily manifest in the following scenarios: i) scenes featuring only partial human bodies, such as arm or leg, which introduces challenges for person detection; and ii) chaotic scenes teeming with people, which causes occlusion and difficulties in identifying interactions. Despite these challenges, DIFFUSIONHOI has shown remarkable improvement over existing approaches. Additionally, the patterns of these failure cases provide valuable insights for future research.

## A.3  Broader Impact

On the positive side, this work proposes a more powerful solution for recognizing complex interactions between humans and objects in a scene. It is particularly effective for few- and zero-shot setups that are common in real-world scenarios. This advancement can significantly contribute to a range of applications, such as healthcare and assistive systems, smart homes and IoT (Internet of Things), security, and more. However, there are also potential negative aspects. Our method carries the risk of being used for continuous monitoring, which could raise concerns over intrusive surveillance and the unauthorized collection of personal data. Additionally, the inversion-based modeling strategy for adjusting diffusion models could be misused to create harmful or false information about individuals. Hence, it is essential to rigorously consider ethical standards and legal regulations to address privacy concerns, so as to avoid potential negative societal impacts.

# B  Detailed Comparison

We present a more detailed comparison of DIFFUSIONHOI with other methods in Table S1. As demonstrated, DIFFUSIONHOI consistently achieves state-of-the-art performance.

# C  Qualitative Results for Image Generation

We present the qualitative results of *relation-driven* sample generation in Fig. S2. It can be observed that the synthesized images exhibit realistic shapes and textures for individual instances, as well as coherent structure and layout for entire scenes. Additionally, the generated images accurately depict the interactions between humans and objects, demonstrating the effectiveness of our proposed inversion-based HOI modeling strategies.

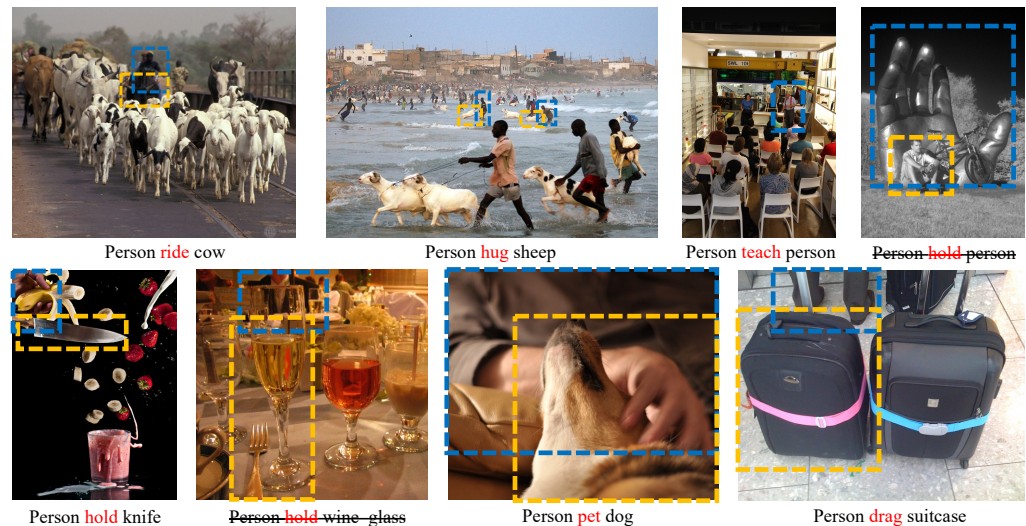

| Person ride cow | Person hug sheep | Person teach person | ~~Person~~ hold ~~person~~ |

| Person hold knife | ~~Person hold wine_glass~~ | Person pet dog | Person drag suitcase |

Figure S1: Typical failure case on HICO-DET[19]. Actions highlighted in red indicate missing predictions that should be detected, while text with ~~strikethrough~~ means wrong predictions that should be removed.

# D License

The V-COCO[21] and SWiG-HOI[22] datasets are released under the MIT license. The HICO-DET[20] dataset is released under the CC0: Public Domain license. The weight of Stable Diffusion is released under CreativeML Open RAIL M License.

Table S1: Detailed comparison for regular HOI detection on HICO-DET[20] and V-COCO[21].

| Method | Backbone | VL Pretrain | Default | | | Known Object | | | V-COCO | |
|---|---|---|---|---|---|---|---|---|---|---|
| | | | Full | Rare | Non-Rare | Full | Rare | Non-Rare | $AP_{role}^{S1}$ | $AP_{role}^{S2}$ |
| iCAN[83] [BMVC18] | R50 | - | 14.84 | 10.45 | 16.150 | 16.26 | 11.33 | 17.73 | 45.3 | - |
| UnionDet[23] [ECCV20] | R50 | - | 17.58 | 11.72 | 19.33 | 19.76 | 14.68 | 21.27 | 47.5 | 56.2 |
| DRG[105] [ECCV20] | R50-FPN | - | 19.26 | 17.74 | 19.71 | 23.40 | 21.75 | 23.89 | 51.0 | - |
| PPDM[24] [CVPR20] | HG104 | - | 21.73 | 13.78 | 24.10 | 24.58 | 16.65 | 26.84 | - | - |
| HOTR[41] [CVPR21] | R50 | - | 23.46 | 16.21 | 25.60 | - | - | - | 55.2 | 64.4 |
| ConsNet[106] [MM20] | R50 | - | 24.39 | 17.10 | 26.56 | - | - | - | - | - |
| AS-Net[40] [CVPR21] | R50 | - | 28.87 | 24.25 | 30.25 | 31.74 | 27.07 | 33.14 | 53.9 | - |
| QPIC[42] [CVPR21] | R50 | - | 29.07 | 21.85 | 31.23 | 31.68 | 24.14 | 33.93 | 58.8 | 61.0 |
| CDN[44] [NeurIPS21] | R50 | - | 31.78 | 27.55 | 33.05 | 34.53 | 29.73 | 35.96 | 62.3 | 64.4 |
| CPChoi[84] [CVPR22] | R50 | - | 29.63 | 23.14 | 31.57 | - | - | - | 63.1 | 65.4 |
| MSTR[107] [CVPR22] | R50 | - | 31.17 | 25.31 | 32.92 | 34.02 | 28.83 | 35.57 | 62.0 | 65.2 |
| UPT[86] [CVPR22] | R50 | - | 31.66 | 25.94 | 33.36 | 35.05 | 29.27 | 36.77 | 59.0 | 64.5 |
| STIP[85] [CVPR22] | R50 | - | 32.22 | 28.15 | 33.43 | 35.29 | 31.43 | 36.45 | 66.0 | 70.7 |
| IF-HOI[108] [CVPR22] | R50 | - | 33.51 | 30.30 | 34.46 | 36.28 | 33.16 | 37.21 | 63.0 | 65.2 |
| ODM[109] [ECCV22] | R50-FPN | - | 31.65 | 24.95 | 33.65 | - | - | - | - | - |
| Iwin[87] [ECCV22] | R50-FPN | - | 32.03 | 27.62 | 34.14 | 35.17 | 28.79 | 35.91 | 60.5 | - |
| MCPC[88] [ECCV22] | R50 | - | 35.15 | 33.71 | 35.58 | 37.56 | 35.87 | 38.06 | 63.0 | 65.1 |
| PViC[89] [ICCV23] | R50 | - | 34.69 | 32.14 | 35.45 | 38.14 | 35.38 | 38.97 | 62.8 | 67.8 |
| PViC†[89] [ICCV23] | Swin-L | - | 44.32 | 44.61 | 44.24 | 47.81 | 48.38 | 47.64 | 64.1 | 70.2 |
| CTAN[50] [CVPR22] | R50 | CLIP | 31.71 | 24.82 | 33.77 | 33.96 | 26.37 | 36.23 | 60.1 | |
| SSRT[47] [CVPR22] | R50 | CLIP | 30.36 | 25.42 | 31.83 | - | - | - | 63.7 | 65.9 |
| DOQ[49] [CVPR22] | R50 | CLIP | 33.28 | 29.19 | 34.50 | - | - | - | 63.5 | - |
| GEN-VLK[3] [CVPR22] | R50 | CLIP | 33.75 | 29.25 | 35.10 | 37.80 | 34.76 | 38.71 | 62.4 | 64.4 |
| HOICLIP[19] [CVPR23] | R50 | CLIP | 34.69 | 31.12 | 35.74 | 37.61 | 34.47 | 38.54 | 63.5 | 64.8 |
| CQL[90] [CVPR23] | R50 | CLIP | 35.36 | 32.97 | 36.07 | 38.43 | 34.85 | 39.50 | 66.4 | 69.2 |
| ViPLO[91] [CVPR23] | ViT-B | CLIP | 37.22 | 35.45 | 37.75 | 40.61 | 38.82 | 41.15 | 62.2 | 68.0 |
| AGER[92] [ICCV23] | R50 | CLIP | 36.75 | 33.53 | 37.71 | 39.84 | 35.58 | 40.2 | 65.7 | 69.7 |
| RmLR[93] [ICCV23] | R50 | MobileBERT | 36.93 | 29.03 | 39.29 | 38.29 | 31.41 | 40.3 | 63.8 | 69.8 |
| ADA-CM[94] [ICCV23] | ViT-L | CLIP | 38.40 | 37.52 | 38.66 | - | - | - | 58.6 | 64.0 |
| DIFFUSIONHOI | VQGAN | Stable Diffusion | **38.12** | **38.93** | **37.84** | **40.93** | **42.87** | **40.04** | **66.8** | **70.9** |
| DIFFUSIONHOI | ViT-L | Stable unCLIP | **42.54** | **42.95** | **42.35** | **44.91** | **45.18** | **44.83** | **67.1** | **71.1** |

†: Models built upon advanced object dector, i.e., $\mathcal{H}$-Deform-DETR[95].

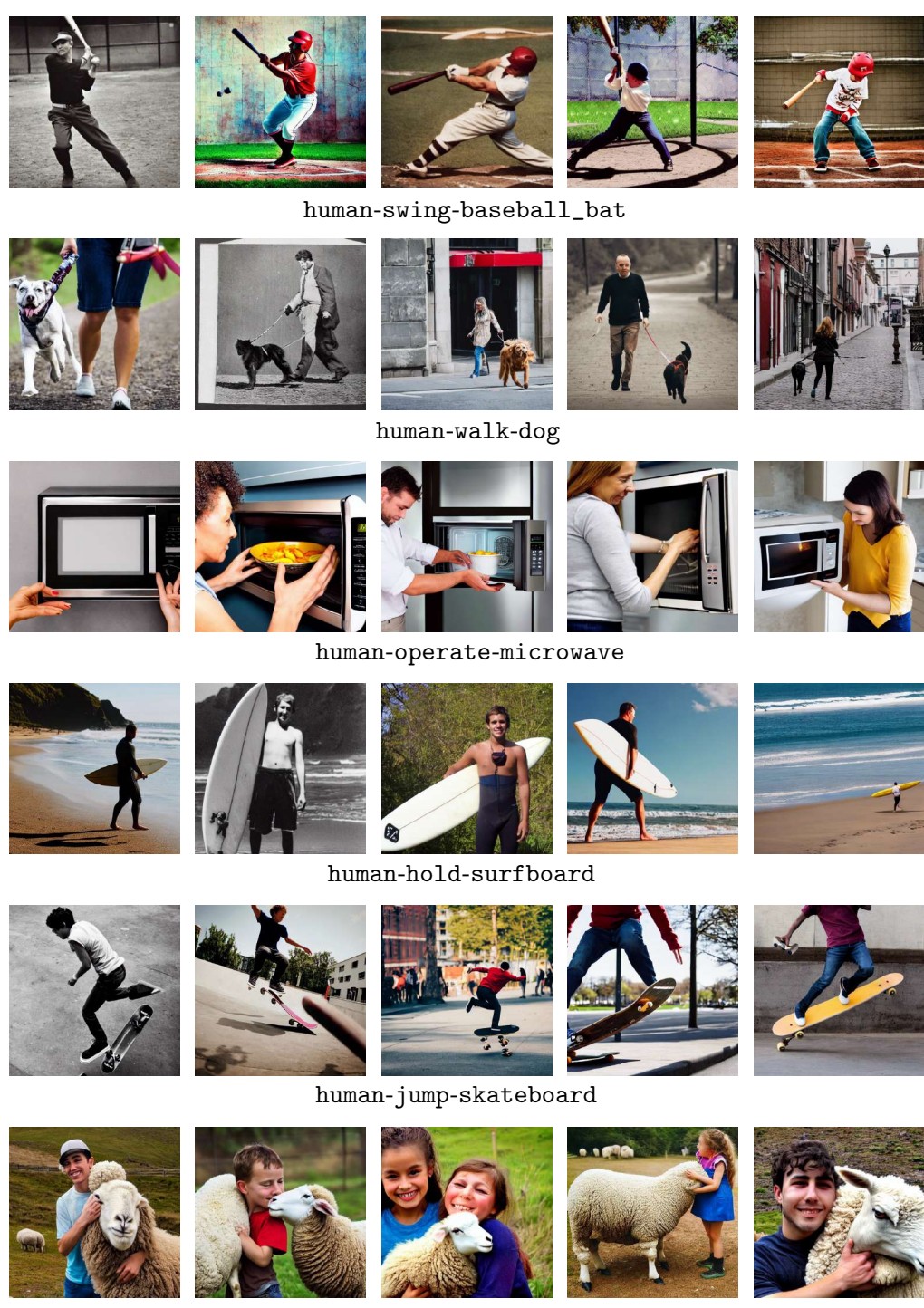

`human-swing-baseball_bat`

`human-walk-dog`

`human-operate-microwave`

`human-hold-surfboard`

`human-jump-skateboard`

`human-hug-sheep`

Figure S2: Qualitative results for relation-driven image generation.

