# OpenReview forum: "Human-Object Interaction Detection Collaborated with Large Relation-driven Diffusion Models"
_NeurIPS.cc/2024/Conference — NeurIPS 2024 poster_

### Official Review · Reviewer_Xw3E · 2024-07-11

**Soundness:** 3
**Presentation:** 3
**Contribution:** 3
**Rating:** 5
**Confidence:** 3

**Summary:**

This paper introduces DIFFUSIONHOI, a novel Human-Object Interaction (HOI) detector that utilizes text-to-image diffusion models for HOI detection. It efficiently focuses on complex relationships between objects, providing a strong basis for HOI modeling. The relation-driven approach enhances image generation capabilities for HOI detection, enriching training samples for rare interactions. Additionally, it improves detector flexibility and accuracy, achieving good performance on HICO-DET and V-COCO benchmarks.

**Strengths:**

- The experiments are comprehensive.
- The design of Inversion-Based HOI Modeling and Relation-Driven Sample Generation are intriguing.-
- Achieving zero-shot generalization on SWiG-DET is impressive.

**Weaknesses:**

- Despite having fewer trainable parameters overall, DIFFUSIONHOI has a large total parameter count, which raises concerns about unfair comparisons with existing methods.
- The authors claim to set a new state-of-the-art on HICO-DET, but the reviewer found that there are better models available, as also mentioned in the paper [79].
- Quantitative results and analysis of failure cases are needed.

[79] Frederic Z Zhang, Yuhui Yuan, Dylan Campbell, Zhuoyao Zhong, and Stephen Gould. Exploring predicate visual context in detecting of human-object interactions. In ICCV, 2023

**Questions:**

N/A

**Limitations:**

Yes

---

> ### Author Rebuttal · Authors · 2024-08-06
>
> **Q1: Concerns about unfair comparisons as DIFFUSIONHOI has a larger total parameter count.**
>
> **A1:** Though DIFFUSIONHOI contains a larger count of parameters, this does not incur heavy consumption in computation resources compared to existing work.
> First, the training time ($\textit{i.e.}$, 5.7 hours for relation embedding learning and 11.5 hours for HOI detection learning) is more efficient than most existing work, which can be evidenced by the comparison of training time below. Second, as the discussion provided in L321-330 and the results summarized in Table 4, the inference time of DIFFUSIONHOI is faster than the two-stage HOI detectors, and comparable to the one-stage HOI detectors without a significant gap.
> Based on the analysis above, we believe we render a fair comparison to existing work.
>
> |Method|Time (Hour)|
> |:-|:-|
> |CDN[39]|25.2|
> |HOTR[36]|23.6|
> |UPT[76]|17.9|
> |STIP[77]|16.4|
> |GEN-VLKT[2]|28.4|
> |HOICLIP[18]|29.1|
> |CQL[80]|29.7|
> |DIFFUSIONHOI|$\textbf{5.7+11.5}$|
>
> More importantly, as stated in L75-77, one valuable contribution of this work is the effective knowledge transfer from diffusion models without heavy fine-tuning. We address this challenge by learning task-oriented embeddings ($\textit{i.e.}$, relational embeddings) as textual prompts, and further transfer the knowledge from relation-driven sample generation and feature extraction two aspects.
> Considering the current trend towards leveraging large-scale pre-trained models to assist in various downstream perception tasks, we believe this work could offer valuable insights to the border community, and thus the larger count of parameters cannot be considered as the drawback of our work.
>
> ---
>
> **Q2: Comparsion to [79].**
>
> **A2:** We provide the comparison between DIFFUSIONHOI with VQGAN and PViC with ResNet-50 as the backbone in Table 1.
> For PViC with Swin-L as the backbone, it leverages $\mathcal{H}$-Deform-DETR as the detector which achieves 48.7 mAP on MS COCO by running merely 12 epochs, significantly higher than DETR which achieves 36.2 mAP by running 50 epochs.
> For a fair comparison, we i) reimplement PCiV with ViT-L as the backbone and DETR as the detector, and ii) augment the two parallel decoders for instance and interaction detection of DIFFUSIONHOI with the technology used in $\mathcal{H}$-Deform-DETR, resulting in DIFFUSIONHOI with the decoder holds a similar detection ability to $\mathcal{H}$-Deform-DETR. The detailed results are summarized below.
>
> |Model| Backbone| Detector| Full|Rare|Non-rare|AP$^{S1}_{role}$|AP$^{S2}_{role}$|
> |:-|:-|:-|:-|:-|:-|:-|:-|
> |PViC|ResNet-50|DETR| 34.69 |32.14 |35.45|62.8 |67.8|
> |DIFFUSIONHOI|VQGAN|$\approx$DETR|$\textbf{38.12}$|$\textbf{38.93}$|$\textbf{37.84}$|$\textbf{66.8}$ |$\textbf{70.9}$|
> |PViC|ViT-L|DETR| 39.89| 40.36|39.98|63.4|68.9|
> |DIFFUSIONHOI|ViT-L|$\approx$DETR|$\textbf{42.54}$|$\textbf{42.95}$|$\textbf{42.35}$|$\textbf{67.1}$|$\textbf{71.1}$|
> |PViC|Swin-L|$\mathcal{H}$-Deform-DETR|44.32 |44.61| 44.24|64.1| 70.2|
> |DIFFUSIONHOI |ViT-L|$\approx$$\mathcal{H}$-Deform-DETR|$\textbf{45.46}$|$\textbf{45.78}$|$\textbf{45.37}$|$\textbf{67.9}$|$\textbf{71.5}$|
>
> As seen, for models with similar backbones, DIFFUSIONHOI achieves much better performance. For instance, 38.12 $\textit{vs.}$ 34.69, and 42.54 $\textit{vs.}$ 39.89.
> On this basis, as observed in the last two rows of the table above, though ViT-L is inferior to Swin-L in downstream tasks, DIFFUSIONHOI still surpasses PViC by 1.14 mAP on the Full set of HICO-DET and 3.8/1.3 mAP on the Scenario1/Scenario2 setup of V-COCO, respectively.
>
> Table 1 and $\S$4.2 will be updated accordingly.
>
> ---
>
> **Q3: Quantitative results and analysis of failure cases.**
>
> **A3:** Thank you for pointing this out. We will include the quantitative results regarding HOI detection in the Appendix for detailed analysis. Additional quantitative results for relation-driven sample generation have already been provided in Figure S1.
>
> Regarding failure case analysis, the visualized results are provided in the PDF attached with "global" response.
> After reviewing the predictions delivered by DIFFUSIONHOI, we found that failure cases primarily manifest in the following scenarios: i) scenes featuring only partial human bodies, such as arm or leg, which introduces challenges for person detection; and ii) chaotic scenes teeming with people, which causes occlusion and difficulties in identifying interactions.
> Despite these challenges, our algorithm has shown remarkable improvement over existing approaches. Additionally, the patterns of these failure cases provide valuable insights for future research.
>
> ---
>
> Thank you for your valuable time and constructive feedback. We will revise the main text with discussions and experimental results according to your review. It is welcome if there are any further questions or suggestions.

---

> > ### Comment · Reviewer_Xw3E · 2024-08-13
> >
> > Thanks for the authors' response. The rebuttal has addressed most of my concerns. I keep my positive score.

---

### Official Review · Reviewer_UENt · 2024-07-13

**Soundness:** 3
**Presentation:** 3
**Contribution:** 3
**Rating:** 7
**Confidence:** 4

**Summary:**

This paper introduces DIFFUSIONHOI, a new HOI detector leveraging text-to-image diffusion models. Unlike previous one-stage or two-stage models, diffusion models excel at discerning mid/low-level visual concepts as generative models and possess strong compositionality to handle novel concepts expressed in text inputs. To steer the focus of diffusion models from instance generation to the relationships between humans and objects, this paper exploits textual inversion and devises a human-object relation inversion strategy grounded in the disentanglement of HOI. Furthermore, to transfer extensive knowledge from large-scale diffusion models to assist in recognizing interactions, the paper leverages both text-prompted image generation and conditioned feature extraction capabilities of diffusion models. By embracing text-to-image diffusion models and facilitating relation-driven image generation and prompting, this method demonstrates superior performance.

**Strengths:**

This paper is well-written and easy to follow. It proposes a new diffusion-based solution, DiffusionHOI, for the human-object interaction (HOI) task. Unlike traditional one-stage and two-stage methods, DiffusionHOI benefits from controllable image generation and HOI knowledge transfer from diffusion models. Without making design changes to the HOI decoder, this method achieves superior performance compared to previous state-of-the-art models by deriving HOI-relevant features to assist in HOI detection.

**Weaknesses:**

1. I am curious about the number of parameters in the VQGAN used as the backbone of DiffusionHOI. According to Table 1, the authors use ViT-L as their backbone, which appears to be significantly larger compared to previous works, especially those using the Res50 network. Could the authors provide more results using a smaller backbone, such as ViT-B, to offer a clearer and fairer comparison between DiffusionHOI and previous state-of-the-art models?
2. Another concern is that the improvement primarily stems from the additional knowledge provided by the powerful diffusion model. Could this be considered unfair when comparing it to previous works that do not have this significant benefit? In this case, how does this approach differ from simply adding more data for training? As I am familiar with HOI but not as familiar with diffusion models, could the authors provide more details about the data scale and parameter scale of the diffusion model used in this paper? Additionally, I look forward to feedback from other reviewers on this matter.

**Questions:**

Please refer to weakness.

**Limitations:**

Please refer to weakness.

---

> ### Author Rebuttal · Authors · 2024-08-06
>
> **Q1.1: Number of parameters in the VQGAN.**
>
> **A1:** We only utilize the encoder of VQGAN as the backbone which contains 39.2M parameters and is smaller than that of ResNet-101 (42.8M). To address your concern about unfair comparison with models using ResNet-50 as the backbone, we provide a detailed comparison with top-leading approaches using ResNet-101 below.
>
> |Model| Backbone| Full|AP$^{S1}_{role}$|AP$^{S2}_{role}$|
> |:-|:-|:-|:-|:-|
> |QPIC|ResNet-101|29.90|58.3|60.7|
> |CDN|ResNet-101|32.07|63.91|65.89|
> |UDP|ResNet-101|32.62|61.3|67.1|
> |Iwin|ResNet-101|32.79|60.9|-|
> |GEN-VLKT|ResNet-101|34.95|63.6|66.0|
> |RmLR|ResNet-101|37.41|64.2|70.2|
> |DIFFUSIONHOI|VQGAN|$\textbf{38.12}$|$\textbf{66.8}$ |$\textbf{70.9}$|
>
> As seen, our DIFFUSIONHOI achieves SOTA performance on both HICO-DET and V-COCO. This verifies the superiority of our proposed method. Table 1 will be updated accordingly.
>
> ---
>
> **Q1.2: Comparison with large backbones.**
>
> **A2:** The implementation of DIFFUSIONHOI relies on pre-trained Diffusion models that do not provide small backbones such as ViT-B. Therefore, we provide a detailed comparison with models using large backbones like ViT-L below to ensure a fair comparison.
>
> |Model| Backbone| Full|AP$^{S1}_{role}$|AP$^{S2}_{role}$|
> |:-|:-|:-|:-|:-|
> |ADA-CM|ViT-L|38.40|58.6|64.0|
> |PViC|ViT-L|39.89|63.4|68.9|
> |DIFFUSIONHOI|ViT-L|$\textbf{42.54}$|$\textbf{67.1}$| $\textbf{71.1}$|
>
> Currently, only a few models provide results on large backbones such as ViT-L, and DIFFUSIONHOI can achieve the best performance on two datasets.
>
> ---
>
> **Q2.1: Unfair comparison to previous work without using additional knowledge.**
>
> **A2.1:** Though prior work does not utilize diffusion models for knowledge transfer, they adopted other visual-linguistic models like CLIP ($\textit{e.g.}$, CTAN [45], SSRT [42], DOQ [44], GEN-VLK [2], HOICLIP [18], CQL [80], AGER[82], RmLR[83], and ADA-CM[84]). We provide a detailed comparison with approaches using pre-training models such as CLIP or MobileBERT in Table 1. It can be seen that when provided with additional knowledge, DIFFUSIONHOI can achieve a clear gap in performance ($\textit{e.g.}$, 38.12 mAP $\textit{vs.}$ 33.75 mAP for GEN-VLK[2], 35.35 mAP for CQL[80], 36.75 mAP for AGER[82], and 36.93 mAP for RmLR[83]), which highlights the effectiveness of our design.
>
> **Q2.2: The difference from methods simply adding more data for training.**
>
> **A2.2:** The primary distinction lies in three aspects:
>
> First, adding training data from other datasets is impractical in real-world applications, as it requires significant time and labor to gather annotated data, which include bounding boxes, object classes, and the interactions between humans and objects. This also imposes significant challenges in generating HOI samples using diffusion models due to the complexity of annotations.
>
> Second, the training data in our work is generated with respect to the relations between humans and objects, resulting in images that depict specific human-object interactions more accurately ($\S$3.3 Relation-Driven Sample Generation). As outlined in L178-180, for a text prompt such as "The man at bat readies to swing at the pitch," the action word "swing" is replaced with the relation embedding learned through relation-centric inversion. This steers diffusion models to focus on complex relationships rather than single objects, without heavy retraining.
>
> Third, as stated in L53-61, we achieve knowledge transfer from diffusion models in two aspects. In addition to data generation, features distinguished for each interaction can be derived with learned relational embeddings as prompts ($\S$3.4 Relation-Inspired HOI Detection). Therefore, we enhance HOI detection with diffusion models in both training data and hypothesis set ($\textit{i.e.}$, feature extraction), while adding more data can only affect the training set.
>
> Additional experiments on HICO-DET are given below to verify the effectiveness of our design. As seen, simply adding more data from SWiG-DET for training (row #2) delivers limited improvement compared to using additional data synthesized with relation-driven sample generation (row #3), or features extracted with relational embeddings as prompts (row #4).
>
> |Method|Full|Rare| Non-Rare|
> |:-|:-|:-|:-|
> |BASELINE|33.24|30.25|34.32|
> |+ data from SWiG-DET only |33.89 |31.13 |34.72 |
> |+ Synthesized Data only|35.49 |36.27| 35.02|
> |+ Relation Prompting only|36.45|35.78|36.71|
> |DIFFUSIONHOI|$\textbf{38.12}$|$\textbf{38.93}$|$\textbf{37.84}$|
>
> **Q2.3: More details about the data and parameter scale of the diffusion model used.**
>
> **A2.3:** As specified in L254, DIFFUSIONHOI is built upon Stable Diffusion v1.5 pre-trained on laion2B-en, a dataset containing 2.3B samples, and subsequently fine-tuned on high-quality subsets of laion2B-en. The total parameter size is 1066M. Despite the large parameter size, as summarized in Table 4, the number of trainable parameters is significantly smaller compared to existing works ($\textit{e.g.}$, 27.6M $\textit{vs.}$ 50.4M for STIP [77], 41.9M for QPIC [37], and 42.8M for GEN-VLKT [2]), and the inference speed is comparable to leading counterparts ($\textit{e.g.}$, 9.49 FPS $\textit{vs.}$ 6.23 for iCAN [74], 7.12 for STIP [77], 3.24 for ADA-CM [84]). This indicates that the usage of diffusion models does not impose a heavy burden on our approach.
>
> Moreover, as stated in L76-77 and L365-366, this work presents an efficient way to transfer knowledge from diffusion models, with human-object relational embeddings learned via relation-centric inversion. This can offer valuable insights to the community, and further advance the exploration of large visual-linguistic pre-training models for downstream tasks.
>
> ---
>
> Thank you for providing such a comprehensive and detailed review. We are committed to revising the experiment section in accordance with your valuable suggestions. Please do not hesitate to post comments if there are any further questions we can help.

---

### Official Review · Reviewer_MQdd · 2024-07-16

**Soundness:** 2
**Presentation:** 3
**Contribution:** 2
**Rating:** 5
**Confidence:** 4

**Summary:**

This paper tackles the human-object interaction (HOI) detection task.

It aims to utilize the feature of generative models like diffusion models to help human-object interaction classification.

More specifically, it utilize the inversion process in the diffusion model to learn the embedding for human-object interaction.

The proposed method obtains obvious improvement over existing methods.

**Strengths:**

- The improvement on performance is significant compared with the baseline.
- The figures are illustrative and helpful for understanding this paper.
- The writing and overall demonstration is good. The problems to tackle are clearly explained, for example, in line 39-61. This does affect my rating.
- The ablation study is though and well organized.

**Weaknesses:**

- The cost of training with the inversion process. As the inversion process is computationally costly and slow, it could affect the training speed. I would like to see more clarification or analysis over the cost of the proposed method.
- The proposed model does not achieves SOTA as it claims. For example, it falls behind PViC (iccv23) with swin-L backbone by a large margin. As the proposed method have a ViT-L backbone, I think it is not fair to not mentioning PViC-swin-L and some other methods. Besides, PViC utilize no external knowledge like CLIP or SD while this method does.
- The general technical design is complicated. This is not a big problem and does not affect my rating.

**Questions:**

- The analysis of the cost.

I will look into the rebuttal and adjust my rating accordingly.

**Limitations:**

Discussion over the limitation is provided.

---

> ### Author Rebuttal · Authors · 2024-08-06
>
> **Q1: Analysis on the training cost.**
>
> **A1:** For relation-centric inversion, unlike the original textual inversion technology that learns text embeddings within the image space, we optimize relation embeddings within the latent space by reconstructing interaction features. This results in reduced training costs. Consequently, the 117 relation embeddings in HICO-DET can be learned within 5.7 hours (23 minutes per relation embedding), using 8 Tesla A40 cards.
>
> |Method| Time (Minute)|
> |:-|:-|
> |Textual Inversion|32|
> |Relation-centric Inversion|23|
>
> For the main training of HOI detection on HICO-DET, as shown in Table 4, our method utilizes significantly fewer trainable parameters compared to existing work ($\textit{e.g.}$, 27.6M $\textit{vs.}$ 50.4M for STIP [77], 41.9M for QPIC [37], and 42.8M for GEN-VLKT [2]), and therefore the training process is completed in just 11.5 hours.
> The comparison of the whole training time with some representative work is summarized below, with all experiments conducted on 8 Tesla A40 cards. It can be observed that our method is more efficient.
>
> |Method|Time (Hour)|
> |:-|:-|
> |CDN[39]|25.2|
> |HOTR[36]|23.6|
> |UPT[76]|17.9|
> |STIP[77]|16.4|
> |GEN-VLKT[2]|28.4|
> |HOICLIP[18]|29.1|
> |CQL[80]|29.7|
> |DIFFUSIONHOI|$\textbf{5.7+11.5}$|
>
> The discussion and results above will be integrated into $\S$4.2.
>
> ---
>
> **Q2: Comparison to SOTA work.**
>
> **A2:** To solidly address your concern, we provide a comprehensive comparison with PViC below. Here PViC with Swin-L leverages $\mathcal{H}$-Deform-DETR as the detector which achieves 48.7 mAP on MS COCO by running merely 12 epochs, significantly higher than DETR which achieves 36.2 mAP by running 50 epochs.
> For a fair comparison, we i) reimplement PViC with ViT-L as the backbone and DETR as the detector, and ii) augment the two parallel decoders for instance and interaction detection of DIFFUSIONHOI with the technology used in $\mathcal{H}$-Deform-DETR, resulting in DIFFUSIONHOI with the decoder holds a similar detection ability to $\mathcal{H}$-Deform-DETR. The detailed results are summarized below.
>
> |Model| Backbone| Detector| Full|Rare|Non-rare|AP$^{S1}_{role}$|AP$^{S2}_{role}$|
> |:-|:-|:-|:-|:-|:-|:-|:-|
> |PViC|ResNet-50|DETR| 34.69 |32.14 |35.45|62.8 |67.8|
> |DIFFUSIONHOI|VQGAN|$\approx$DETR|$\textbf{38.12}$|$\textbf{38.93}$|$\textbf{37.84}$|$\textbf{66.8}$ |$\textbf{70.9}$|
> |PViC|ViT-L|DETR| 39.89| 40.36|39.98|63.4|68.9|
> |DIFFUSIONHOI|ViT-L|$\approx$DETR|$\textbf{42.54}$|$\textbf{42.95}$|$\textbf{42.35}$|$\textbf{67.1}$|$\textbf{71.1}$|
> |PViC|Swin-L|$\mathcal{H}$-Deform-DETR|44.32 |44.61| 44.24|64.1| 70.2|
> |DIFFUSIONHOI |ViT-L|$\approx$$\mathcal{H}$-Deform-DETR|$\textbf{45.46}$|$\textbf{45.78}$|$\textbf{45.37}$|$\textbf{67.9}$|$\textbf{71.5}$|
>
> As seen, with a similar backbone ($\textit{i.e.}$, ResNet-50 and VQGAN), our method surpasses PViC by 3.43 mAP on HICO-DET, 4.0 mAP on V-COCO. Next, after reimplementing PViC with DETR and ViT-L as the backbone, the performance is still inferior to DIFFUSIONHOI ($\textit{i.e.}$, 42.54 $\textit{vs.}$ 39.89 on HICO-DET). Finally, after augmenting DIFFUSIONHOI with tricks used in $\mathcal{H}$-Deform-DETR, the performance boosts to 45.46 mAP on HICO-DET. Please note that ViT-L (employed by DIFFUSIONHOI) is inferior to Swin-L (employed by PViC) for downstream tasks.
>
> We will provide a discussion in the experiment section, to highlight methods not using additional knowledge ($\textit{e.g.}$, PViC) during comparison.
>
> ---
>
> **Q3: Complicated technical design.**
>
> **A3:** This study represents the first attempt to utilize diffusion models for HOI detection, from the perspective of both data generation and conditioned feature extraction. This may bring a fundamental shift in architectural design and training strategy.
> However, we agree that simplicity in design is crucial for reimplementation and extension by the community. We will definitely consider this as our future work.
>
> ---
>
> Thank you so much for your careful review and suggestive comments. We will update our manuscript with the training time analysis and a more detailed comparison with the SOTA approaches. Please feel free to post your feedback if you have any further questions.

---

### Author Rebuttal · Authors · 2024-08-06

**To all reviewers**

We express our sincere gratitude to all reviewers for their valuable time and thorough assessment of our manuscript. In response, we have carefully addressed each concern raised, and provided point-to-point clarifications which shall be integrated into the new version of our manuscript.

We are gratified by the positive feedback from all reviewers, particularly regarding the significant improvement in performance, the quality of presentation, and the comprehensiveness of our experiments.

Foremost among the concerns is the comparison to top-leading approaches. We provide additional experiments to ensure an all-inclusive comparison, of which our method can still achieve the best performance.

Additionally, Reviewer MQdd raises concerns about the training efficiency of DIFFUSIONHOI. We appreciate this feedback and have included a detailed analysis regarding training costs to clarify this issue.
Similarly, Reviewer UENt and Reviewer Xw3E highlight concerns about the parameter count of our approach. In response, we have provided additional experiments and discussions to clarify the motivation and contribution of using diffusion models for downstream tasks.

Other questions, such as the complexity of the technical design of our approach, the differences from simply adding training data, more details about the diffusion model used in this work, and analysis on failure cases, have also been addressed accordingly.

---

### Comment · Area_Chair_2zLJ · 2024-08-09

Hi, All,
Please make the best use of the discussion period to learn better from each other's ideas. Note that the discussion ends on Aug 13. Thank you.

Best,
AC

---

### Decision · Program_Chairs · 2024-09-25

**Decision:**

Accept (poster)

**Comment:**

The paper introduces DIFFUSIONHOI, a novel method for Human-Object Interaction (HOI) detection utilizing text-to-image diffusion models to improve visual cue recognition and compositional reasoning. The approach demonstrates significant performance improvements across multiple datasets, including HICO-DET and V-COCO. Despite concerns about training efficiency, parameter count, and technical complexity, the authors provided comprehensive analyses and additional comparisons, reinforcing the method's effectiveness. Reviewers acknowledged the innovative integration of diffusion models and clear presentation, although they noted the architecture's complexity and suggested further exploration of failure cases. Overall, the paper is recommended for acceptance.